# Pulse Oximetry and Congenital Heart Disease Screening: Results of the First Pilot Study in Morocco

**DOI:** 10.3390/ijns6030053

**Published:** 2020-06-30

**Authors:** Nadia El Idrissi Slitine, Fatiha Bennaoui, Craig A. Sable, Gerard R. Martin, Lisa A. Hom, Amal Fadel, Soufiane Moussaoui, Nadir Inajjarne, Drissi Boumzebra, Youssef Mouaffak, Said Younous, Lahcen Boukhanni, Fadl Mrabih Rabou Maoulainine

**Affiliations:** 1Neonatal Intensive Care Unit, Mother and Child Hospital, Mohammed VI University Hospital, Marrakech 40000, Morocco; fatihabennaoui@yahoo.fr (F.B.); amalfadfad@gmail.com (A.F.); soufiane.lueur@gmail.com (S.M.); nadir.inajjarne.s@gmail.com (N.I.); fadl2020@hotmail.com (F.M.R.M.); 2Laboratory Childhood, Health and Development, Marrakesh Medical School, Cadi Ayyad University, Marrakech 40000, Morocco; ymouaffak@yahoo.fr (Y.M.); anesthesie.marrakech@gmail.com (S.Y.); 3Children’s National Heart Institute, George Washington University School of Medicine, Washington, DC 20010, USA; csable@childrensnational.org (C.A.S.); gmartin@childrensnational.org (G.R.M.); lhom@childrensnational.org (L.A.H.); 4Cardio-Vascular Surgery Center, Arazzi Hospital, Mohammed VI University Hospital, Marrakech 40000, Morocco; boumzebradriss@hotmail.com; 5Pediatric Intensive Care, Mother and Child Hospital, Mohammed VI University Hospital, Marrakech 40000, Morocco; 6Gyneco-Obstetrical Unit, Mother and Child Hospital, Mohammed VI University Hospital, Marrakech 40000, Morocco; boukhanni.lahcen@gmail.com

**Keywords:** neonatal screening, congenital heart disease, pulse oximetry, telemedicine, Mohammed VI Hospital, Marrakesh, Morocco

## Abstract

Congenital heart disease (CHD) is the most common congenital malformation. Diagnosis of critical congenital heart disease (CCHD), the most severe type of congenital heart disease, in a newborn may be difficult. The addition of CCHD screening, using pulse oximetry, to clinical assessment significantly improves the rate of detection. We conducted a pilot study in Morocco on screening neonates for critical congenital heart disease. This study was conducted in the maternity ward of Mohammed VI University Hospital of Marrakesh, Morocco, and included asymptomatic newborns delivered between March 2019 and January 2020. The screening of CCHD was performed by pulse oximetry measuring the pre- and post-ductal saturation. Screening was performed on 8013/10,451 (76.7%) asymptomatic newborns. According to the algorithm, 7998 cases passed the screening test (99.82%), including one inconclusive test that was repeated an hour later and was normal. Fifteen newborns failed the screening test (0.18%): five CCHD, five false positives, and five CHD but non-critical. One false negative case was diagnosed at 2 months of age. Our results encourage us to strengthen screening for CCHD by adding pulse oximetry to the routine newborn screening panel.

## 1. Introduction

The Kingdom of Morocco has 36 million inhabitants and Marrakesh is the third largest city in Morocco; the natality rate in 2019 was 16.9% and the neonatal mortality was 16‰.

Congenital heart disease (CHD) is the most common congenital malformation, with an incidence of 8 per 1000 live births. Critical congenital heart disease (CCHD), defined as CHD requiring early detection and surgical intervention or cardiac catheterization in the first year of life to sustain life, occurs in 2.5 to 3 per 1000 live births. These cardiac lesions are responsible for approximately 40% of deaths due to congenital birth defects in the first year of life [1]. There is no national published data on the prevalence of the congenital heart disease or on the mortality that it can cause. Current screening techniques do not detect all cases. Prenatal ultrasound in some studies still detects less than 50% of the cases of CCHD [2]. Signs and symptoms suggestive of CCHD are not always present initially in the first few days of life, so physical examination also does not always identify neonates with CCHD [3]. The late management of a child with multiple visceral failure worsens the prognosis of heart disease. Delayed or missed diagnosis of CCHD in asymptomatic newborns may result in significant morbidity and mortality [4].

In 2011, the American Academy of Pediatrics (AAP) and the American Heart Association (AHA) endorsed pulse oximetry screening for the early detection of CCHD to prevent subsequent morbidity and mortality that may occur because of late diagnosis [5]. Several large population-based studies have recently shown that pulse oximetry, coupled with existing screening techniques, can increase the detection rate of CCHD to more than 90% [2,3,6,7]. 

In Morocco, screening for CCHD is still not implemented and there are no publications on this subject. This work is a pilot study conducted with the following objectives:To study the feasibility of CCHD screening in our context.To improve the early detection of CCHD for better management.To improve the timely detection of neonates with other causes of hypoxemia.

And a long-term objective: to implement the screening of CCHD in newborn delivery hospitals in Morocco, and to integrate it into the routine examination of newborns, before their discharge from the hospital, according to international recommendations on best practice.

## 2. Material and Methods

This prospective study was conducted from March 2019 to January 2020 in the maternity ward of the Mother and Child Hospital, at the Mohammed VI University Hospital of Marrakesh in Morocco. This study evaluated neonates who were normal according to standard neonatal examinations. The study protocol was approved on 9 January 2019 by the research ethics committee of Marrakesh University Medical School, Input and guidance for the implementation study were provided by the Children’s National Hospital in Washington, DC, USA, with whom we have been working in partnership since 2010. The patients diagnosed with CCHD were discussed in telemedicine meetings with the Children’s National team.

Newborns with symptoms suggestive of congenital heart disease or any other disease, or those who had an antenatal diagnosis of CHD, and newborns lost to follow up were excluded from the study.

Informed consent was obtained orally from the parents, after explaining the value of the screening and before collecting pulse oximetry data during the routine examination of the newborn. The screening was performed by residents and medical students. No additional staff was employed to perform the test. According to local recommendations, the screening test was performed before discharge—for vaginal delivery before 24 h and for C section after 24 h—except during the weekend days or during holidays, where the test could not be performed because of the lack of the medical staff to do so.

Two oximeters designed specifically for congenital heart disease screening were used: MASIMO^®^ RAD-97 and RAD-7. The CCHD screening was done according to the AAP algorithm (4).

If the newborn failed the test, he or she was examined by a pediatric cardiologist, who performed a cardiac ultrasound urgently. When a CCHD was diagnosed, the file was saved for multidisciplinary meetings between the local staff and the Children’s National Staff using videoconferencing.

## 3. Results

The total number of newborns in the maternity ward of Mohammed VI Marrakesh University Hospital during this study was 10,451; 8013 newborns were screened and 2438 newborns missed the screening (Figure 1). The sex ratio was 1.02, with 4166 boys (52.2%) and 3830 girls (47.8%) screened.

We had an equal distribution of newborns of rural origin 51% (4086 cases) and urban origin in our study, 329 cases (4.1%) were from gestational diabetic mothers, 247 of the mothers (3.08%) had preeclampsia, and 40 of them had other chronic pathologies (0.5%) (e.g., dysthyroidism, psychiatric pathology). 

Medication usage during pregnancy occurred in 841 cases (10.48%) of mothers; iron supplementation was used in 5570 (69.4%), insulin in 801 (10%), nifedipine in 577 (7.2%), and in 1477 (18.4%), other medications, such as thyroid supplementation or vitamins. The average maternal age was 26.95 years, with a minimum age of 16 years and a maximum of 47 years, 321 of mothers (4%) were between 35 and 40 years old. A family history of congenital heart disease was present in 190 cases (2.37%). Among the newborns tested positive, one case also had an undocumented past medical history of congenital heart disease in the family. The parents were consanguineous in 1037 cases (12.93%), and none of the screened newborns were carriers of multiple malformation syndrome.

The prevalence of prematurity was 8.6%, the average gestational age was 38 weeks (with an extreme range of 33 and 42 weeks), the average birth weight was 3186 g, intrauterine growth restriction (IUGR) represented 0.5% (40 newborn) and macrosomia 2.28% (including 6 positive screenings), and twin pregnancy represented 162 cases (2%) of the cases.

According to the algorithm, 7998 newborns (99.82%) passed the test and 15 cases failed the screening test: 0.18% (Figure 1). The test was performed ≤24 h in 7051 cases (88%), 5128 of the newborns (64%) were screened between 12 and 24 h of life, and 1923 newborns (24%) were screened before 12 h of life. The test was performed >24 h of life in 962 newborns (12%). The screening was well-accepted by the parents; none of them refused it.

Among the 15 (1.8/1000) newborns who failed the screening test, 5 (0.6/1000) had CCHD, 5 (0.6/1000) were false positives and 5 (0.6/1000) had CHD but non-critical. One false negative case was identified in this study; a child who presented at 2 months of age in the emergency department for dyspnea and was diagnosed with coarctation.

No antenatal cases were diagnosed in this study, six of the positive screened newborns (40%) were infants of diabetic mothers, and three newborns with failed tests were excluded from the study because they did not attend the cardio pediatric consultation.

The echocardiography performed in the 15 newborns who failed the screening test showed the following results (Table 1):5 CCHD: 1 case of D-transposition of great arteries, 1 case of double outlet right ventricle (DORV), transposition of great arteries, and pulmonary stenosis, 1 case of coarctation, and 2 cases of hypoplastic left heart syndrome (HLHS).5 with other heart diseases: 1 case of AV canal, 1 case of large ASD, 2 cases of hypertrophic cardiomyopathy, and 1 case of single atrium.5 false positives: 1 persistent pulmonary hypertension (PPHN), 2 sepsis, and 2 normal heart.1 false negative: coarctation of aorta at 2 months of age.

## 4. Discussion

Pulse oximetry screening increased the detection of CCHD at the Mohammed VI University Hospital of Marrakesh. Morocco is a North African country and is characterized by its biodiversity, higher risk of consanguinity in rural areas, and that it does not have a well-established program for antenatal diagnosis of CCHD which increases the importance of timely diagnosis of CCHD in the newborn period. The oximetry screening test was easy, simple, reliable, reproducible, acceptable, discriminating, well-accepted by parents and caregivers, and did not involve parental anxiety.

Positive CCHD screens occurred in 1.8/1000 (0.18%) of newborns in this study. This rate of positive screening was less than reported in Switzerland (0.74%) [7], China (0.43%) [6], and the United Kingdom (0.98%) [8], but comparable to a rate in India (0.34%) [9].

Pulse oximetry has a good specificity and sensitivity and thereby fulfills the criteria for mass screening [10]. Additionally, most of the data in the literature suggest a favorable cost-effectiveness of this technique [2,3,10,11,12]. One of the major limitations of this study was that sensitivity and specificity could not be commented on like did other studies. Some newborns that screened positive were lost to follow-up and did not show up for the cardio-pediatric consultation, which was scheduled as an emergency. Hence, there were some logistical difficulties in collecting surveillance data on newborns that were discharged from the hospital without having this screening test, particularly those born on holidays or weekends, or false negative infants who passed the test.

To the best of our knowledge, this is the first published study on the use of pre- and post-ductal pulse oximetry as a screening test for the diagnosis of CCHD in Morocco, or North Africa. In Abu Dhabi, the Health Authority Abu Dhabi (HAAD), in collaboration with the Children’s National Medical Center (Children’s National), successfully implemented CCHD screening at the emirate level using a “train-the-trainer”, two-tiered approach, starting with two pilot hospitals then rolling the program out to all birthing facilities [13]. In Tunisia, a study based on the post ductal screening showed very low incidence of CHD and did not mention the CCHD [14].

The screening of CCHD after 24 h has low false negative rates. In this study, there was a tendency to discharge apparently healthy newborns before 24 h. The length of stay in the maternities of Marrakesh University Hospital after delivery was very short: <36 h for the C section and < 24 h for the normal deliveries. Therefore, waiting after 24 h to perform a screening test was impracticable in the majority of the cases. The AAP recommends that all asymptomatic neonates be screened for critical CHD between 24 and 48 h of age or prior to discharge [6,8].

Screening with pulse oximetry also detected other causes of hypoxemia. It allowed a timely diagnosis and adequate management for respiratory distress syndrome, pulmonary hypertension or sepsis. It therefore contributes to the early detection of these potentially fatal but curable pathologies [15]. In this study, two cases of sepsis were diagnosed and treated early, with good outcomes, and one case of PPHN and two newborns had normal findings.

Limitations in this screening study included the failure to detect one infant with coarctation of the aorta, who was diagnosed at 2 months of age when the duct closed and signs of heart failure were noticed. Left heart obstructive conditions have been well documented to have lower sensitivity. An additional limitation was that not all the neonates were screened, because of the lack of equipment and lack of physician staff that could perform the screening, especially during the weekend and holidays. The nurses and midwives were not involved in this project for several reasons: there were not enough human resources to do the screening and it was not considered to be in the scope of their practice [16].

Investigation of positive screens is difficult to perform in settings where echocardiography is not immediately available. Often, the lack of echocardiography is linked to lack of pediatric cardiologists on site at maternity hospitals. This problem can be solved by implementing video conferences with maternity hospitals and local tertiary care centers where a pediatric cardiologist is available.

These results encourage us to strengthen screening for heart disease by examining the family history of newborns and educating the medical and paramedical community about the benefits of this test. In fact, our experience has shown a very high death rate after the surgery of TGA, in part because most present at 2 to 6 weeks of life when the risk of surgery is much higher. The TGA newborn that was screened in this study was successfully operated on. Certainly, the maturation of surgery programs in Morocco are at the point when early diagnosis can change the surgical risk profile; therefore, the timing of pulse screening is critical.

Screening of CCHD should be part of the newborn screening program that is planned to be implemented in Morocco, including blood spot testing and hearing screening, that are not yet established. Efforts to improve the access to cardiovascular surgery and to develop intensive care expertise are also a big challenge in Morocco, and will contribute to improve the outcomes of CCHD bedside screening.

## 5. Conclusions

Screening for CCHD is a reliable method for the early detection of critical congenital heart disease and even non-cardiac conditions. We think that it will have positive repercussions on infant mortality and morbidity in Morocco. It is an optimal test and it adapts perfectly to our context. We hope to implement it locally and nationally, as consistent with international best practices for newborn screening, to allow for timely detection of the infants born with CCHD in Morocco.

## Figures and Tables

**Figure 1 IJNS-06-00053-f001:**
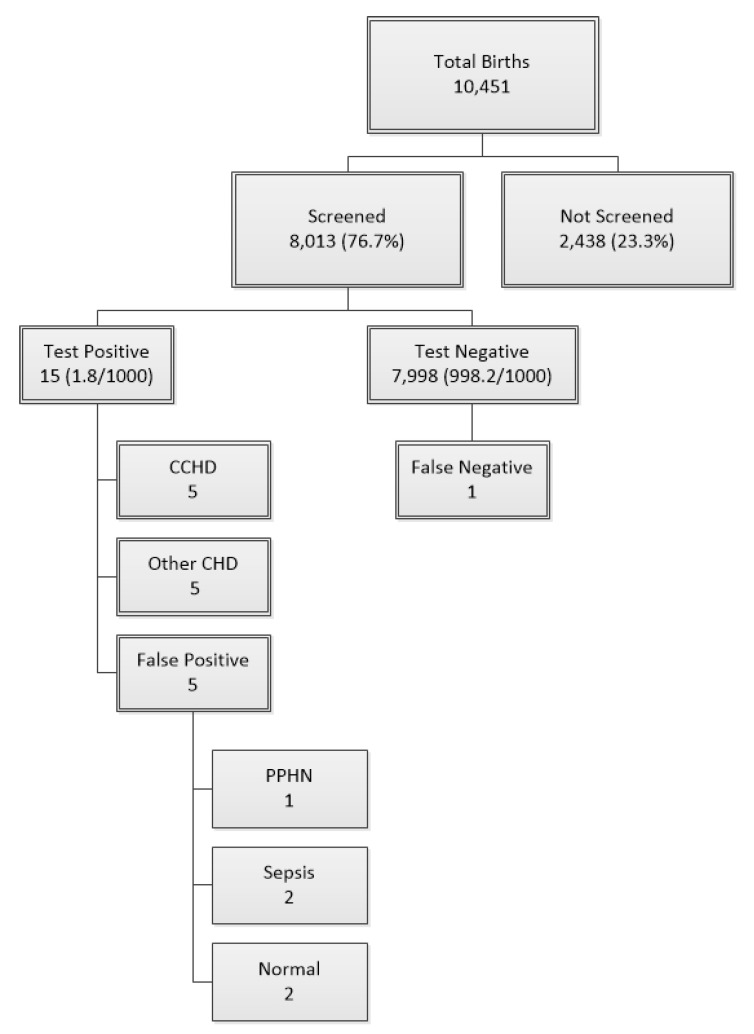
Distribution of the newborns enrolled in the study. 10,451 newborns were enrolled in the study among them 8013 newborns were screened for CCHD, and 2438 newborns were not tested. 15 babies (1.8/1000) failed the screening (test positive): among them, 5 CCHD were diagnosed, 5 non-critical CHD, and 5 false positives: 1 PPHN, 2 sepsis, and 2 normal hearts. CCHD: critical congenital heart disease; CHD: congenital heart disease; PPHN: persistent pulmonary hypertension of the newborn.

**Table 1 IJNS-06-00053-t001:** Repartition of failing test newborns.

Diagnosis	Number of Cases	Outcome
Critical Congenital Heart Disease		
D-Transposition of great arteries	1	Operated with good results
Double outlet right ventricle, transposition of great arteries, and pulmonary stenosis	1	Stable, waiting for surgery
Coarctation and Persistent ductus arteriosus	1	Waiting for surgery
Hypoplastic left heart syndrome	2	Died
Non-Critical Congenital Heart Diseases		
Single atrium.	1	Stable, waiting for surgery
Hypertrophic myocardium	2	Normal heart
Large atrial septal defect	1	Stable, waiting for surgery
Atrio ventricular canal defect	1	Stable, waiting for surgery
False positive		
Persistent pulmonary hypertension of the newborn	1	Normal
Sepsis	2	Recovered
Normal	2	Normal
False negative		
Coarctation of aorta	1	Stable, waiting for surgery

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
