# Peer review of "Pulse Oximetry and Congenital Heart Disease Screening: Results of the First Pilot Study in Morocco"

_2409-515X, 2020, doi:10.3390/ijns6030053_

Round 1

Reviewer 1 Report

Thanks you for asking me to review this paper which is the first report of pulse oximetry screening (POS) in Morrocco.

The authors should be congratulated on this piece of work which demonstrates the importance and feasibility of POS in this setting.

I have a few comments

Main points

Some explanation or acknowledgement of the number of unscreened babies (almost 25%). Why were they missed? What measures will be introduced to correct this?

Algorithm should be explained in full

Timing of screening generally <24 hours but FP rate was not high. This would be worth discussing.

Minor points

There are some phrases where the English could be improved and it would be worth a native English speaker reviewing the manuscript.

Specifically 'multiple visceral', 'poly malformatif' 'escaped' (should be 'missed' or 'not screened')

Author Response

To the reviewer n°1

I thank you for your remarks and  tried to respond point by point

  • Some explanation or acknowledgement of the number of unscreened babies (almost 25%). Why were they missed? What measures will be introduced to correct this
  • Response: as I explained in the article, we had a high rate of unscreened babies mostly due to the unavailability of the screening team during the weekends, the holidays, and some  other days (exam days for example) and during the time of  the study there were a lot of these missing days of the screening. We should do our best to involve the nurses and midwives into practicing CHD screening which is a new practice for them not considered in the scope of their activities.
  • Algorithm should be explained in full
  • Response: the algorithm was fully explained

10,451 newborn were enrolled in the study, among them 8,013 newborns were screened for CCHD, and 2438 newborns were not tested. 15 babies (1.8 /1000) failed the screening (test positive), among them 5 CCHD were diagnosed, 5 non-critical CHD and 5 false positive: 1 PPHN, 2 sepsis and 2 normal hearts.

  • Timing of screening generally <24 hours but FP rate was not high. This would be worth discussing.
  • Response: it is well known that if we perform the test before 24 hours, the risk of FP rate will be high, in this study it is not the case, I don’t have any explanation, we should continue and enlarge the study to have more results.
  • There are some phrases where the English could be improved and it would be worth a native English speaker reviewing the manuscript Specifically 'multiple visceral', 'poly malformatif' 'escaped' (should be 'missed' or 'not screened')
  • Response: it was done: polymalformatif syndrome was replaced by multiple malformation syndrome

Reviewer 2 Report

IJNS--822325 

TITLE: Pulse Oximetry and Congenital Heart Disease Screening: Where Are We in Morocco?

The study was not conducted to answer the question of “Where Are We in Morocco? It is a pilot study in Marrakesh. Maybe, the title could be changed to Pulse Oximetry and Congenital Heart Disease Screening: Results of the First Pilot Study in Morocco or similar.

ABSTRACT

Well written. No comments or suggestions.

INTRODUCTION

Lines 33-41: Authors present “global” data. It would be useful if they can consider to make an introductory comment about the situation in Morocco and/or in Marrakesh, partially addressed later on line 50.

Including in the introduction (briefly) some demographic facts on the Kingdom of Morocco (inhabitants, yearly births, neonatal and/or infant mortality) and Marrakesh (i.e.: the fourth largest city), would very useful for the reader. (Some from lines 129-133).

Suggest that lines 55 and 56-58 are separated from lines 52-54.

MATERIAL AND METHODS

Why was the period March 2019 to January 2020 chosen?

Line 74: at what postnatal age is the screening supposed to be performed? (On line 108 of results there are details about post-natal age, but a brief sentence should be added here on what the local protocol calls for).

Line 79-80: I suggest that the concept is “softened”. Not every baby with positive screening test needs a cardiac US urgently.

Is PI registered during the SpO2 screening test?

RESULTS

On line 85: instead of a comma (,) use a semicolon in …. this study was 10,451; 8013 newborns …

I suggest that line 91 is moved to a later paragraph where the test positive infants are reported (to around lines 112-114). Furthermore, there is no need to write “Note” in results. Suggestion: Six of the positive screened newborns (40%) were infants of diabetic mothers.

Similarly, for line 96 (to around lines 112-114).

Figure 1. Suggest % are added to the numbers in all “categories”.

Line 116: no need to write “Note”.

Line 126: Was the PI registered in this infant? Normal? (Maybe a comment can be entered on line 166-167 of discussion.

DISCUSSION

Minor issues:

Lines 153-156: pretty long sentence, with several punctuation marks. It could be separated in 2 or more sentences to improve readability.

Line 157: the post-natal age when the screening test was performed belongs in results. Additionally, the AAP recommendation can be supplemented by different recommendations (i.e.: Ewer’s)

Line 161-162: Has to be re-written (…..other causes of hypoxemia and this timely diagnosis and ensured adequate management ….)

Line 169-170: Authors can consider adding a reference recently published in this same Journal that describes similar difficulties and challenges in developing nations in Latin America. (Int. J. Neonatal Screen. 2020, 6(1),21; https://doi.org/10.3390/ijns6010021).

Line 180: is this referring to the local previous experience at the Mohammed VI University Hospital of Marrakesh?

Line 186: suggestion: ... including blood spot testing and hearing screening that are not yet established.

Author Response

Response to the reviewer n°2

Thankyou for these interesting remarks

Comments and Suggestions for Authors

IJNS--822325 

  • TITLE: Pulse Oximetry and Congenital Heart Disease Screening: Where Are We in Morocco?

The study was not conducted to answer the question of “Where Are We in Morocco? It is a pilot study in Marrakesh. Maybe, the title could be changed to Pulse Oximetry and Congenital Heart Disease Screening: Results of the First Pilot Study in Morocco or similar.

Response: the title was changed to Pulse Oximetry and Congenital Heart Disease Screening: Results of the First Pilot Study in Morocco as you suggested

ABSTRACT

Well written. No comments or suggestions.

INTRODUCTION

Lines 33-41: Authors present “global” data. It would be useful if they can consider to make an introductory comment about the situation in Morocco and/or in Marrakesh, partially addressed later on line 50.

Including in the introduction (briefly) some demographic facts on the Kingdom of Morocco (inhabitants, yearly births, neonatal and/or infant mortality) and Marrakesh (i.e.: the fourth largest city), would very useful for the reader. (Some from lines 129-133).

Response: The Kingdom of Morocco accounts 36 million of inhabitants, the natality rate in 2019 is 16.9 %, the neonatal mortality is 16‰ and Marrakesh is the third largest city.

Congenital heart disease (CHD) is the most common congenital malformation, with an incidence of 8 per 1000 live births. Critical congenital heart disease (CCHD), defined as CHD requiring early detection and surgical intervention or cardiac catheterization in the first year of life to sustain life, occurs in 2.5 to 3 per 1,000 live births. These cardiac lesions are responsible for approximately 40% of deaths due to congenital birth defects in the first year of life [1]. There is no national published data on the prevalence of the congenital heart disease or on the mortality that it can cause.

Suggest that lines 55 and 56-58 are separated from lines 52-54.

Response: This work is a pilot study conducted with the following objectives:

  • To study the feasibility of CCHD screening in our context.
  • To improve the early detection of CCHD for better management.
  • To improve the timely detection of neonates with other causes of hypoxemia.

And a long-term objective: to implement the screening of CCHD in newborn delivery hospitals in Morocco, and to integrate it into the routine examination of the newborn, before his discharge from the hospital, according to international recommendations on best practice.

MATERIAL AND METHODS

Why was the period March 2019 to January 2020 chosen?

Response: we started the study at this date of march and gave our preliminary results until January but we will definitely extend the study to get more results

Line 74: at what postnatal age is the screening supposed to be performed? (On line 108 of results there are details about post-natal age, but a brief sentence should be added here on what the local protocol calls for).

Response:According to local recommendations, the screening test was performed before discharge, for vaginal delivery before 24 hours and for C section after 24 hours, the test couldn’t be done during the weekend days or during holidays, because of the lack of the medical staff to perform it.

Line 79-80: I suggest that the concept is “softened”. Not every baby with positive screening test needs a cardiac US urgently.

Response: In the Moroccan context, if we don’t perform the cardiac US urgently, the babies can be discharged before 24 hours and they may not come back to the hospital.

Is PI registered during the SpO2 screening test? response: yes

RESULTS

On line 85: instead of a comma (,) use a semicolon in …. this study was 10,451; 8013 newborns …

response: it’s done

I suggest that line 91 is moved to a later paragraph where the test positive infants are reported (to around lines 112-114). Furthermore, there is no need to write “Note” in results. Suggestion: Six of the positive screened newborns (40%) were infants of diabetic mothers.

It’s done

Similarly, for line 96 (to around lines 112-114).

It’s done

Figure 1. Suggest % are added to the numbers in all “categories”.

Line 116: no need to write “Note”.

Line 126: Was the PI registered in this infant? Normal? (Maybe a comment can be entered on line 166-167 of discussion. Limitations in this screening study included the failure to detect one infant with coarctation of the aorta, who was diagnosed at 2 months of age when the duct closed and the signs of heart failure were noticed.

DISCUSSION

Minor issues:

Lines 153-156: pretty long sentence, with several punctuation marks. It could be separated in 2 or more sentences to improve readability.

In this study, there was a tendency to discharge apparently healthy newborns before 24 hours. The length of stay in the maternities of Marrakesh University Hospital after delivery was very short: <36 h for the C section, and < 24 h for the normal deliveries. Therefore, waiting after 24 hours to perform a screening test was impracticable in the majority of the cases.

Line 157: the post-natal age when the screening test was performed belongs in results. Additionally, the AAP recommendation can be supplemented by different recommendations (i.e.: Ewer’s)

The time of the screening was < 12 hours after birth in 24% of the neonates, 12-24 hours after birth in 64%, and > 24 hours after birth in 12%.

The AAP recommends that all asymptomatic neonates be screened for critical CHD between 24 and 48 hours of age or prior to discharge [6,8].

Line 161-162: Has to be re-written (…..other causes of hypoxemia and this timely diagnosis and ensured adequate management ….)

Response: Screening with pulse oximetry also detected other causes of hypoxemia. It allowed a timely diagnosis and adequate management for respiratory distress syndrome, pulmonary hypertension or sepsis.

Line 169-170: Authors can consider adding a reference recently published in this same Journal that describes similar difficulties and challenges in developing nations in Latin America. (Int. J. Neonatal Screen. 2020, 6(1),21; https://doi.org/10.3390/ijns6010021).

  1. Sola A, Rodríguez S, Young A, Lemus Varela, and al. CCHD Screening Implementation Efforts in Latin American Countries by the Ibero American Society of Neonatology (SIBEN).  J. Neonatal Screen.2020, 6(1), 21.

Line 180: is this referring to the local previous experience at the Mohammed VI University Hospital of Marrakesh? Response: Yes

Line 186: suggestion: ... including blood spot testing and hearing screening that are not yet established.

Response: it was done
